# Therapeutic Plasma Exchange in Acute Liver Failure: A Real-World Study in Mexico

**DOI:** 10.3390/healthcare13162059

**Published:** 2025-08-20

**Authors:** Jose Carlos Gasca-Aldama, Jesús Enrique Castrejón-Sánchez, Mario A. Carrasco Flores, Enzo Vásquez-Jiménez, Paulina Carpinteyro-Espin, Juanita Pérez-Escobar, Karlos Dhamian Gutierrez-Toledo, Pablo E. Galindo, Marcos Vidals-Sanchez, Paula Costa-Urrutia

**Affiliations:** 1Department of Intensive Care Unit, Hospital Juárez de Mexico, Ciudad de Mexico 07760, Mexicomacf_fcam@hotmail.com (M.A.C.F.); dr_kdgt@hotmail.com (K.D.G.-T.); concord_markv@hotmail.com (M.V.-S.); 2Department of Nephrology, Hospital Juárez de Mexico, Ciudad de Mexico 07760, Mexico; enzo.vas.ji@gmail.com; 3Transplant Department, Hospital Juárez de Mexico, Ciudad de Mexico 07760, Mexico; paucarpi@gmail.com (P.C.-E.); dra.jsperez@gmail.com (J.P.-E.); 4Department of Nephrology, Hospital General de Mexico, Ciudad de Mexico 06720, Mexico; galindozip@gmail.com; 5Medical Affairs, Terumo BCT, Edificio Think MVD, Montevideo 11300, Uruguay; paula.costa@terumobct.com

**Keywords:** acute liver failure, plasma exchange, high-volume therapeutic plasma exchange, standard-volume therapeutic plasma exchange, intensive care unit

## Abstract

**Background/Objectives**: Acute liver failure (ALF) is a life-threatening condition with high mortality in nontransplant candidates. Therapeutic plasma exchange (TPE) has emerged as a promising intervention for removing inflammatory mediators and toxic metabolites. In Latin America, data on the efficacy of TPE in ALF patients are limited. This real-world study aimed to compare 30-day survival outcomes between patients receiving standard medical treatment (SMT) and those receiving SMT plus TPE. **Methods:** We analyzed 25 ALF patients admitted to the tertiary intensive care unit (ICU) of Hospital Juárez of Mexico City, Mexico, from 2018 to 2024. Patients received either standard medical treatment (SMT group, n = 12) or SMT with TPE (TPE group, n = 13), including high-volume TPE (n = 8) and standard-volume TPE (n = 5). Survival analysis was performed via Kaplan–Meier estimates, and binomial regression analysis was run to estimate the mortality probability stratified by the hepatic encephalopathy grade. Results: At 30 days, survival was significantly greater in the TPE group (92%) than in the SMT group (50%) (*p* = 0.02). The greatest survival benefit was observed in patients with Grade 4 encephalopathy. The ICU stay was longer in the TPE group, reflecting the complexity of ALF management. **Conclusions:** TPE significantly improves 30-day survival in ALF patients compared with SMT alone, supporting its role as an adjunct therapy. Further studies are needed to refine patient selection and optimize treatment protocols.

## 1. Introduction

Acute liver failure (ALF) is a life-threatening condition characterized by rapid deterioration of hepatic function, severe coagulopathy, and hepatic encephalopathy in patients without pre-existing liver disease. The most common etiologies are viral hepatitis and drug-induced liver injury (DILI), mainly acetaminophen toxicity [1].

The etiology of ALF is a key factor in determining the prognosis and treatment strategy and varies by region. In North America, Japan, and Europe, the leading causes in adults include DILI, viral hepatitis, and cryptogenic liver failure of unknown origin (indeterminate ALF) [1,2,3]. In contrast, in developing countries, acute viral hepatitis remains the primary cause [4]. The prognosis of ALF is highly variable, with survival rates largely dependent on timely supportive care and, in many cases, liver transplantation. However, limited organ availability poses a significant challenge, highlighting the need for alternative therapeutic strategies to improve survival in nontransplant candidates [5].

In Latin America, regional differences further shape the etiology and management of ALF. In Brazil, viral hepatitis, particularly hepatitis A virus (HAV), is the predominant cause of ALF among children, accounting for up to 82.6% of pediatric cases [6]. This high burden has made HAV a major public health concern. In Colombia, liver transplantation is a central intervention in both acute and chronic liver failure, with alcoholic cirrhosis noted as a primary underlying etiology. Notably, post-transplant survival rates in Colombian cohorts reach 82% at one year and 72% at five years [7]. In Argentina, a multicenter study involving 154 adult patients with ALF reported that hepatitis B and autoimmune hepatitis were the most frequent etiologies, with no cases of acetaminophen overdose observed [8].

In Mexico, ALF remains a critical health concern. From 1998 to 2009, there were 2193 reported ALF-related deaths, with the mortality rate increasing from 13.1 to 40.2 deaths per 10 million inhabitants during this period. This increasing trend underscores the growing impact of ALF on the Mexican population [9].

Standard medical treatment (SMT) for ALF primarily focuses on supportive measures, including infection control, the management of cerebral edema, and hemodynamic stabilization. However, given the systemic inflammatory response and multiorgan dysfunction frequently observed in ALF, additional interventions targeting the underlying pathophysiology are needed [10,11]. Therapeutic plasma exchange (TPE) has emerged as a promising extracorporeal therapy, facilitating the removal of inflammatory mediators, toxic metabolites, and coagulation disturbances associated with ALF [12,13].

A pivotal randomized controlled trial by Larsen et al. [14] demonstrated significantly improved transplant-free survival with high-volume TPE (HV-TPE) (58.7% vs. 47.8%) compared with SMT in patients with ALF. Nevertheless, higher volumes of plasma exchange may increase the risk of transfusion-associated complications, including volume overload, which can exacerbate cerebral edema. In this context, a randomized open-label controlled study conducted by Maiwall et al. [15] revealed that standard-volume TPE (SV-TPE) is safe and effective and improves survival, possibly by mitigating cytokine storms and reducing ammonia levels.

Despite accumulating evidence supporting the use of TPE in ALF patients, data specific to Latin America remain scarce [16]. The objective of this study is to evaluate and compare 30-day survival outcomes in patients with acute liver failure (ALF) treated with standard medical treatment (SMT) alone versus those treated with SMT combined with therapeutic plasma exchange (TPE), including both high-volume and standard-volume protocols, in a tertiary intensive care unit in Mexico.

## 2. Materials and Methods

### 2.1. Study Design and Population

This was a single-center, retrospective observational study conducted at the intensive care unit (ICU) of Hospital Juárez in Mexico City. The study analyzed real-world clinical data from 25 patients diagnosed with acute liver failure (ALF) between 2018 and 2024. Patients were divided into two sequential cohorts: 12 patients who received standard medical treatment (SMT) alone (treated between 2018 and 2021) and 13 who received SMT combined with therapeutic plasma exchange (TPE) (treated between 2021 and 2024). Among the 13 patients in the TPE group, 8 received HV-TPE, whereas the remaining 5 patients received SV-TPE. This non-randomized design reflects the real-world implementation of TPE once it became available in the institution.

We confirm that this study was conducted in accordance with the ethical principles outlined in the Declaration of Helsinki (1975), as revised in 2013. Ethical approval was obtained from the Institutional Review Board of Hospital Juárez, Mexico, on 12 October 2023 (approval reference number HJM 071/24-R). Written informed consent was obtained from all participants for inclusion in the study and for the publication of this manuscript.

### 2.2. Eligibility

Eligible patients were male or female, aged 18 years or older, and had a confirmed diagnosis of ALF irrespective of the underlying etiology. Laboratory and imaging studies were performed to confirm the diagnosis of ALF and exclude other causes. These included coagulation parameters (PT and INR), liver function tests (ASAT, ALAT, bilirubin, and albumin), renal function, serum ammonia, and viral serologies for hepatotropic viruses (HAV, HBV, HCV, and HEV). Additional tests included autoimmune markers, toxicology screening (including acetaminophen levels), and pregnancy testing when appropriate. Imaging (ultrasound and/or angiography) was used to exclude structural liver disease and Budd–Chiari syndrome.

Exclusion criteria included the following: (1) pregnant patients, (2) patients in the immediate or late postpartum period, (3) patients with acute-on-chronic liver failure, (4) patients who withdrew informed consent, (5) patients transferred to another hospital, and (6) patients with incomplete clinical or laboratory data in the study database.

### 2.3. Intervention

The SMT included antibiotics (piperacillin and tazobactam), N-acetylcysteine, and anti-cerebral edema treatment. Carbapenem and antifungal agents (fluconazole) were initiated for patients whose clinical status worsened, with or without elevated procalcitonin.

Vascular access for TPE was achieved via an 11-French, double-lumen hemodialysis catheter in the internal jugular vein. TPE was performed with the Spectra Optia^®^ Apheresis System (Terumo Blood and Cell Technologies, Lakewood, CO, USA), which employs continuous-flow centrifugation for efficient plasma separation.

The total blood volume (TBV) was calculated as 70 mL/kg of the patient’s weight, and the total plasma volume (TPV) was determined via the following formula: TPV = TBV × (1 − hematocrit). Standard-volume TPE was performed by exchanging the patients’ TPV 1.0-fold, whereas HV-TPE targeted 8 to 12 L exchange of patients’ TPV per session. The duration of each TPE session was approximately 3–4 h for SV-TPE and 5–6 h for HV-TPE. 

The anticoagulant used was acid citrate dextrose at a ratio of 1:14. Plasma exchange was performed via the use of 20% human albumin and fresh frozen plasma in a 1:3 ratio as replacement fluids. Neither cryoprecipitate nor cryosupernatant was used, as fresh frozen plasma is the only approved plasma product for TPE at our institution

### 2.4. Clinical and Biochemical Evaluations

Baseline data, including liver enzyme levels (ASAT and ALAT), serum creatinine levels, lactate dehydrogenase levels, serum ammonia levels, the use of vasopressors, bilirubin levels, and serum sodium levels, were collected. The severity of organ dysfunction was assessed via the Sequential Organ Failure Assessment (SOFA) score [17], whereas the Model for End-Stage Liver Disease (MELD) score was used to assess liver disease severity and the need for liver transplantation [18]. Additionally, the King’s College criteria were applied to assess the need for liver transplantation in patients with ALF not associated with acetaminophen toxicity [19], and the Acute Physiology and Chronic Health Evaluation II (APACHE II) score was used to assess disease severity and predict mortality in critically ill patients [20].

### 2.5. Statistical Analysis

For all the statistical analyses, we combined patients treated with HV-TPE and those treated with SV-TPE due to the lack of statistical power to analyze them separately.

The demographic and baseline characteristics of the SMT and TPE groups were compared via appropriate statistical tests. Continuous variables were assessed for normality. Normally distributed variables were compared using one-way ANOVA, while the Kruskal–Wallis test was used for non-normally distributed variables. Categorical variables were compared using Fisher’s exact test or the chi-square test, as appropriate.

Survival data were analyzed via Kaplan–Meier estimates to assess survival probabilities for each treatment group (SMT vs. TPE). Kaplan–Meier curves were generated via the survfit function from the survival package in R [21], with ICU days as the survival time and 30-day mortality as the primary outcome. Differences in survival between groups were compared via the log-rank test, with statistical significance set at *p* < 0.05. Binomial regression analysis was run to estimate the mortality probability stratified by the degree of hepatic encephalopathy. All the statistical analyses were performed via R software v.4.1.0 [22].

## 3. Results

A total of 25 patients with acute liver failure were included: 12 SMT and 13 received SMT plus TPE. At 30 days, survival was 92% (12/13) in the TPE group and 50% (6/12) in the SMT group.

The only non-surviving patient in the TPE group had Grade 4 encephalopathy, a SOFA score of 18, an APACHE II score of 26, and a MELD score of 34, and the etiology of ALF was hepatitis A. This patient, treated with HV-TPE, survived only one day in the ICU. The baseline parameters were comparable between the groups, except for the encephalopathy grade, which was significantly greater in the TPE group (Table 1). Considering all 25 patients, the most prevalent etiology of ALF was hepatitis A (52%, n = 13), followed by non-acetaminophen drug hepatotoxicity (idiosyncratic DILI). No cases of acetaminophen drug hepatotoxicity were reported. Considering etiology by group, hepatitis A was also the most common hepatitis but was significantly more common in the TPE group (69%) than in the SMT group (33%, Table 1).

Kaplan–Meier survival analysis showed a significantly higher 30-day survival rate in the TPE group compared to the SMT group (92% vs. 50%, *p* = 0.02; Figure 1). The SMT group had a sharp decline in survival probability within the first 15 days, while the TPE group maintained a stable survival curve (Figure 1).

While both groups started with similar patient numbers (TPE group = 13, SMT group = 12), the survival probability in the SMT group declined from 0.83 on day 1 to 0.52 by day 3, reaching 0 by day 15. In contrast, the TPE group maintained a high survival probability of 0.92 from days 1 to 30 (95% CI: 0.70–1.00).

Binomial regression analysis shown that TPE treatment was independently associated with reduced 30-day mortality (β = −3.18, SE = 1.43, *p* = 0.03), even after adjusting for encephalopathy grade. Although patients with Grade 4 HE had the highest mortality risk, the protective effect of TPE was observed across all encephalopathy strata. The greatest reduction in mortality was observed in patients with Grade 4 encephalopathy, although this did not reach statistical significance (β = 2.07, SE = 1.69, *p* = 0.22) (Table 2, Figure 2).

The duration of ICU stays varied between the groups. The SMT group typically stayed under five days, whereas the TPE group showed a broader distribution, extending beyond 20 days (Figure 3). Mechanical ventilation was prolonged in both groups, with notable outliers in the TPE group, highlighting the complexity of ALF management with TPE.

## 4. Discussion

The findings of this study revealed a significant survival benefit associated with TPE in ALF patients compared with SMT alone. The Kaplan–Meier survival analysis revealed a statistically significant improvement in 30-day survival rates for patients receiving TPE (*p* = 0.02). This aligns with the literature supporting TPE, both HV-TPE and SV-TPE, as promising adjunct therapies in ALF management, likely due to their role in mitigating systemic inflammation, improving hemodynamic stability, and reducing hepatic encephalopathy severity through the removal of inflammatory mediators and toxic metabolites [14,15,23].

However, our findings contrast with those of Burke et al. [24], who reported no survival benefit of plasma exchange in a multicenter real-world cohort. Although their study included a larger sample size across multiple UK liver transplant centers, it lacked a clearly defined mortality endpoint beyond hospital discharge, making direct comparisons challenging.

Another key difference between the studies is the etiology of ALF. Burke et al. [24] reported that acetaminophen toxicity was the most prevalent cause in the TPE group, whereas in our cohort, hepatitis A was the most common etiology. While acetaminophen toxicity is a leading cause of ALF in North America and Europe, it is less common in developing regions. Given that etiology plays a crucial role in prognosis, further research is needed to better understand TPE outcomes in specific ALF subtypes [1,13].

In the absence of liver transplantation, the prognosis remains poor, particularly in patients with advanced hepatic encephalopathy. Our real-world study suggests that TPE offers a clinically meaningful survival advantage by enhancing hepatic detoxification and modulating immune dysregulation, potentially bridging patients to either spontaneous recovery or transplantation [11,12].

Despite the observed survival benefit, TPE-treated patients had longer ICU stays and greater variability in mechanical ventilation duration. This likely reflects the prolonged supportive care required for ALF recovery and the challenges of managing critically ill patients undergoing extracorporeal therapies. Notably, patients in the TPE group presented with significantly higher grades of hepatic encephalopathy at baseline, on average nearly two grades higher than those in the SMT group (Table 1). This greater severity would typically be associated with worse outcomes, yet the TPE group demonstrated a significantly higher 30-day survival rate. This finding reinforces the potential therapeutic benefit of TPE, even in patients with advanced encephalopathy. It also suggests that the survival difference observed is unlikely to be due to baseline imbalances favoring the TPE group. Additionally, the longer ICU stay observed in the TPE group may, in part, reflect survivor bias, as patients who lived longer had extended hospitalization periods. We addressed this imbalance by performing a binomial regression adjusted for encephalopathy grade, which confirmed that TPE was independently associated with reduced mortality. However, due to the limited sample size, we were unable to perform stratified or matched analyses, which we recognize as a limitation and a recommendation for future studies [22,25].

The strengths of our study include the use of a well-defined ALF cohort, rigorous inclusion criteria, and the application of standardized treatment protocols for both SMT and TPE patients. However, several limitations must be acknowledged, chiefly the small sample size, which reduces statistical power and generalizability. However, it reflects the real-world epidemiology of ALF in our setting, where hepatitis A is the leading cause—unlike in high-income countries, where acetaminophen toxicity is more common. In Latin America, limited access to high-cost liver support systems like MARS or SPAD underscores the value of TPE as a more accessible, cost-effective option. Despite the small cohort, we observed a meaningful reduction in 30-day mortality with TPE.

Given the sample size, we grouped patients receiving high- and standard-volume TPE, based on the recent evidence suggesting SV-TPE may offer similar benefits to HV-TPE with lower costs and fewer complications. Descriptive trends suggest similar outcomes in terms of safety profiles. Further studies with larger cohorts are needed to confirm these observations. While our study demonstrated a survival advantage, it did not clarify the optimal TPE regimen, frequency, or duration required for maximal therapeutic benefit.

In addition, treatment group allocation was sequential: patients in the SMT group were treated between 2018 and 2021, prior to the availability of liver support therapies in our ICU, while TPE became available and was implemented from 2021 onward. This timeline may introduce temporal bias related to changes in ICU practices. However, baseline severity scores (APACHE II, SOFA, and MELD) were comparable between groups, and the TPE group presented with more advanced hepatic encephalopathy at admission, suggesting similar or even higher baseline severity. The longer ICU stay observed in the TPE group may be partially explained by survivor bias. Many patients in the SMT group died within the first few days of admission, whereas most patients in the TPE group survived beyond this early critical phase. Thus, prolonged ICU duration in the TPE group likely reflects longer survival rather than increased disease complexity or resource use.

Another important limitation is the absence of long-term follow-up and quality-of-life outcomes. Our study focused exclusively on 30-day survival, as this reflects the most critical window for mortality in patients with ALF, particularly in non-transplant candidates. However, we acknowledge that assessing long-term survival and functional recovery, including cognitive and physical outcomes, is essential to fully understand the sustained impact of TPE. Future prospective studies should incorporate long-term follow-up and patient-reported outcomes to better evaluate the overall benefit of TPE in this population.

Future research should focus on multicenter, prospective randomized trials with larger patient cohorts to confirm our findings and refine patient selection criteria. As ALF remains a challenging condition with high mortality, integrating TPE into a multimodal treatment approach represents a promising avenue to improve patient outcomes and potentially expand the therapeutic window for liver transplantation [26].

## 5. Conclusions

In conclusion, our study supports the use of TPE as a beneficial intervention in ALF patients, demonstrating a significant survival advantage over the SMT in Mexico. While these findings reinforce the role of TPE in ALF management, further prospective studies are warranted to establish definitive guidelines and optimize therapeutic strategies for this critically ill patient population.

## Figures and Tables

**Figure 1 healthcare-13-02059-f001:**
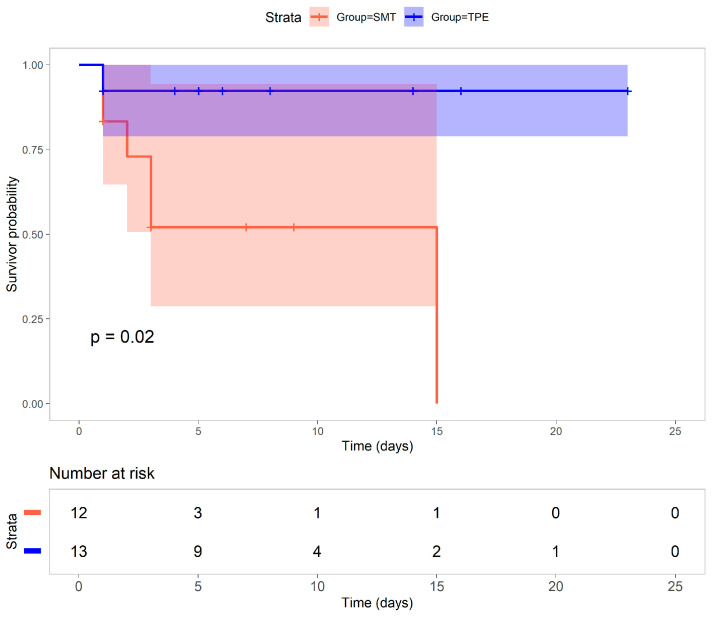
Survival analysis. Kaplan–Meier survival analysis of ALF patients treated with standard medical treatment (SMT) vs. SMT in conjunction with therapeutic plasma exchange (TPE). The log-rank test was used to assess differences between groups. The median survival and 30-day mortality rates are indicated. Statistical significance is reported as *p* values.

**Figure 2 healthcare-13-02059-f002:**
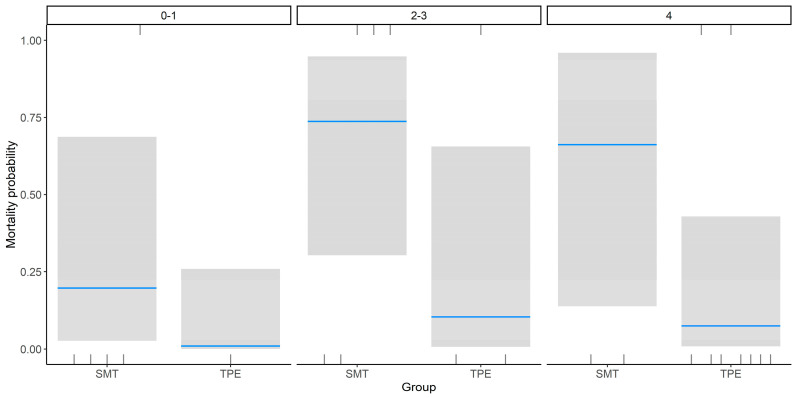
Mortality probability. Binomial regression analysis of ALF patients treated with standard medical treatment (SMT) alone vs. SMT combined with therapeutic plasma exchange (TPE), stratified by encephalopathy grade (0–4).

**Figure 3 healthcare-13-02059-f003:**
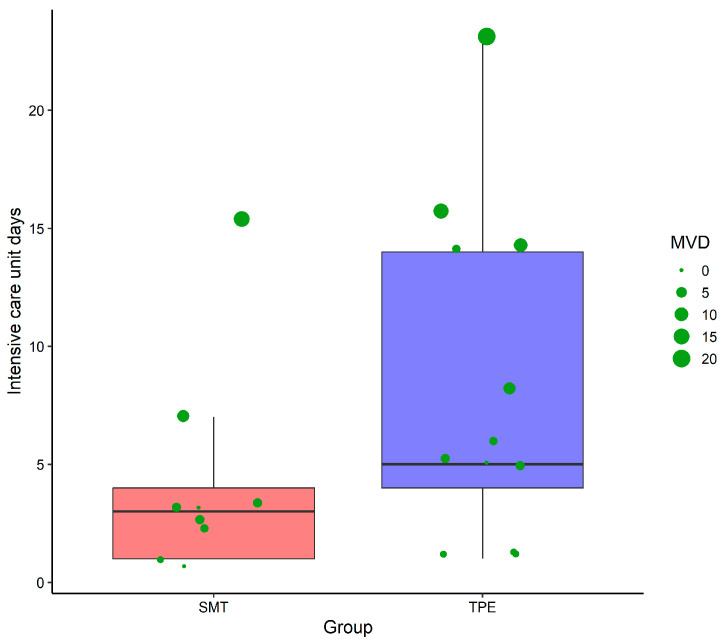
Comparison of intensive care unit (ICU) stay duration and mechanical ventilation days (MVDs) in acute liver failure patients: standard medical treatment (SMT) alone vs. SMT combined with therapeutic plasma exchange (TPE).

**Table 1 healthcare-13-02059-t001:** Baseline demographic and clinical characteristics of patients with acute liver failure in the standard medical treatment (SMT) and therapeutic plasma exchange (TPE) groups. Values are presented as mean ± standard deviation for continuous variables or as percentages for categorical variables.

	SMT (n = 12)	TPE (n = 13)	*p* Value
Gender/female (%)	33	85	0.02
Age	28.3 ± 9.3	29.6 ± 7.8	0.51
Body mass index	27.9 ± 3.9	27.4 ± 3.8	0.71
Etiology (%)			
Hepatitis A	33	69	0.16
DILI	25	8	0.32
Mushroom poisoning	17	8	0.59
Ischemic hepatitis	17	0	0.22
Autoimmune hepatitis	0	8	1.00
ALL	8	0	0.48
AFLP	0	8	1.00
APACHE II	16.6 ± 8.6	18.8 ± 7.4	0.49
Kings College	1.9 ± 1.2	1.8 ± 0.7	0.53
SOFA	7.8 ± 5.0	12.2 ± 5.4	0.11
MELD	34.0 ± 7.2	36.9 ± 7.6	0.35
MELD (%)	59.4 ± 13.9	61.6 ± 8.7	0.87
Hepatic encephalopathy	1.7 ± 1.5	3.4 ± 1.2	0.005
Grade 0–1	5	1	
Grade 2–3	5	3	
Grade 4	2	9	
Creatinine	2.1 ± 1.3	3.1 ± 3.7	0.79
Lactate	6.7 ± 5.0	9.0 ± 6.1	0.35
Bilirubin	13.5 ± 10.5	15.1 ± 9.0	0.25
INR	10.3 ± 6.6	15.6 ± 7.3	0.07
DHL	1078.8 ± 1234.2	1721.8 ± 2092.1	0.51
TGO	2240.7 ± 2880.4	1786.4 ± 1309.9	0.70
TGP	2189.9 ± 2438.8	3678.6 ± 2233.9	0.04
Sodium level	139.0 ± 7.0	135.3 ± 7.0	0.20

Abbreviations: SMT, standard medical treatment; TPE, therapeutic plasma exchange; DILI, drug-induced liver injury; ALL, acute lymphoblastic leukemia; AFLP, acute fatty liver of pregnancy; INR, international normalized ratio; LDH, lactate dehydrogenase; AST (TGO), aspartate aminotransferase; ALT (TGP), alanine aminotransferase; MELD, Model for End-Stage Liver Disease; SOFA, Sequential Organ Failure Assessment; APACHE II, Acute Physiology and Chronic Health Evaluation II.

**Table 2 healthcare-13-02059-t002:** Binomial regression analysis of mortality probability in ALF patients. The model included a treatment group (standard medical treatment and therapeutic plasma exchange (TPE) and a severity of encephalopathy (Grades 2–3 and Grade 4).

	Estimates	Standard Error	*p*-Value
(Intercept)	−1.40	1.12	0.21
Group TPE	−3.18	1.43	0.03
Encephalopathy (Grades 2–3)	2.43 Survival	1.46	0.10
Encephalopathy (Grade 4)	2.07	1.69	0.22

## Data Availability

The data set used and/or analyzed during the current study is available from the corresponding author upon request.

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
