# Peer review of "Therapeutic Plasma Exchange in Acute Liver Failure: A Real-World Study in Mexico"

_healthcare, 2025, doi:10.3390/healthcare13162059_

Round 1
Reviewer 1 Report
Comments and Suggestions for Authors
Major Comments:
- Sample Size and Statistical Power:
- The sample size is small (n=25), which limits the statistical power and the generalizability of the findings.
- The decision to combine HV-TPE and SV-TPE groups is understandable, but the authors should discuss in more depth the clinical implications of potentially different effects between these modalities.
- Selection Bias:
- The treatment allocation was sequential (SMT: 2018–2021; TPE: 2021–2024), which introduces potential temporal and selection biases. It is important to address how evolving ICU practices or changes in patient severity over time may have influenced the outcomes.
- Baseline Imbalances:
- There is a statistically significant difference in hepatic encephalopathy grades between the SMT and TPE groups (p=0.005), with more severe cases in the TPE group. While this would typically bias against the TPE group, it also complicates interpretation. The authors should discuss this thoroughly and consider stratified or matched analyses.
- Survivor Bias:
- Longer ICU stays in the TPE group could reflect survivor bias. This should be explicitly acknowledged and discussed in the limitations.
- Missing Long-Term Outcomes:
- The study focuses on 30-day survival but does not include long-term follow-up or quality-of-life outcomes. These aspects are essential for understanding the true benefit of TPE in ALF.
Minor Comments:
- Language and Grammar:
- Minor grammatical edits are needed throughout. For example, replace “noncetaminophen” with “non-acetaminophen” (line 170).
- Use consistent terminology (e.g., “SMT group” vs. “standard medical treatment group”).
- Figures and Tables:
- Improve clarity of Figure 1 (Kaplan-Meier curve): consider enlarging axis labels and adding a legend to identify groups clearly.
- Include the total number at risk at specific time points in Kaplan-Meier figures.
- Ethics Statement:
- The informed consent statement should be clearly placed in the Methods or Ethics section rather than embedded at the end.
- Conflict of Interest:
- While a potential COI is declared (one author is employed at Terumo BCT), a stronger statement clarifying that Terumo had no role in study design, analysis, or manuscript preparation would improve transparency.
Recommendation:
Major Revision
The study has merit and clinical relevance, particularly for settings with limited transplant access. However, addressing the methodological limitations, clarifying potential biases, and refining the discussion are necessary before the manuscript can be considered for publication.
Comments on the Quality of English Language
- Minor grammatical edits are needed throughout. For example, replace “noncetaminophen” with “non-acetaminophen” (line 170).
- Use consistent terminology (e.g., “SMT group” vs. “standard medical treatment group”).
Author Response
Comment 1. Sample Size and Statistical Power:
- The sample size is small (n=25), which limits the statistical power and the generalizability of the findings.
- The decision to combine HV-TPE and SV-TPE groups is understandable, but the authors should discuss in more depth the clinical implications of potentially different effects between these modalities.
Authors respond: We thank the reviewer for this valuable comment.
We fully acknowledge that the sample size of our study (n=25) is limited, and we now emphasize this more clearly as a study limitation in the Discussion section. However, we believe the cohort reflects the real-world clinical context of acute liver failure (ALF) in our region. Importantly, our data highlight a high prevalence of hepatitis A as the leading etiology of ALF—an epidemiological pattern distinct from that observed in North America and Europe, where acetaminophen toxicity predominates. In this regard, the study provides much-needed local data for Latin America, where access to high-cost liver support systems (e.g., MARS or SPAD) is limited or unavailable. Showing a significant reduction in 30-day mortality, even in a small sample, underscores the potential clinical value of therapeutic plasma exchange (TPE) as a more accessible and cost-effective strategy in resource-constrained settings. We have added this contextual discussion in the revised Discussion section in Limitations of the study.
Regarding the decision to combine high-volume TPE (HV-TPE) and standard-volume TPE (SV-TPE), this was based on two main considerations. First, the primary objective of our study was to evaluate the impact of TPE (regardless of volume) versus standard medical treatment (SMT), not to compare different TPE modalities, given the limited sample size. Second, recent clinical evidence suggests that SV-TPE may provide comparable survival benefits to HV-TPE while reducing treatment-related costs and potential complications associated with high plasma volumes. We have now elaborated on this rationale in the Discussion section (in Limitations of the study) to clarify both the clinical and methodological justifications for this grouping.
We trust this expanded response addresses the reviewer’s concern and adds important context to the interpretation of our findings.
Comment 2. Selection Bias:
- The treatment allocation was sequential (SMT: 2018–2021; TPE: 2021–2024), which introduces potential temporal and selection biases. It is important to address how evolving ICU practices or changes in patient severity over time may have influenced the outcomes.
Author Response:
We appreciate the reviewer’s thoughtful observation. The sequential allocation reflects the fact that between 2018 and 2021, no liver support systems were available in our ICU. Therapeutic plasma exchange (TPE) was introduced as a treatment option starting in 2021. While this design may introduce temporal bias, baseline characteristics—including age, MELD, SOFA, and APACHE II scores—were comparable between groups. Notably, the TPE group had more severe hepatic encephalopathy at admission, suggesting greater baseline severity.
We now acknowledge and discuss these potential sources of bias more explicitly in the revised Discussion section
Comment 3. Baseline Imbalances:
- There is a statistically significant difference in hepatic encephalopathy grades between the SMT and TPE groups (p=0.005), with more severe cases in the TPE group. While this would typically bias against the TPE group, it also complicates interpretation. The authors should discuss this thoroughly and consider stratified or matched analyses.
Author Response. We thank the reviewer for this important observation. Indeed, our analysis showed that the TPE group had significantly more severe hepatic encephalopathy at baseline (p = 0.005). We agree that this imbalance complicates interpretation but also reinforces the robustness of the observed survival benefit in the TPE group. Despite being more severely ill at presentation, patients receiving TPE had significantly better 30-day survival, suggesting a clinically meaningful effect of the intervention.
To further explore this, we performed a binomial regression analysis adjusting for encephalopathy grade, which confirmed a significant reduction in mortality associated with TPE (β = –3.18, SE = 1.43, p = 0.03), independent of encephalopathy severity. These results are now discussed in greater depth in the revised Discussion section, and the potential impact of baseline imbalance is explicitly acknowledged as a study limitation.
Although matched or stratified analyses would have been ideal, the small sample size limited the feasibility of these approaches. We now highlight this limitation in the manuscript and suggest that future studies with larger cohorts consider stratified or propensity-matched designs to better control for baseline imbalances.
Comment 4. Survivor Bias:
- Longer ICU stays in the TPE group could reflect survivor bias. This should be explicitly acknowledged and discussed in the limitations.
Author Response: We agree with the reviewer and appreciate the opportunity to clarify this point. The longer ICU stays observed in the TPE group are indeed likely influenced by survivor bias, as most deaths in the SMT group occurred within the first few days of ICU admission, while the majority of TPE-treated patients survived beyond this critical period. We now explicitly acknowledge and discuss this in the Limitations section of the revised manuscript, noting that extended ICU duration in the TPE group may partly reflect the longer survival time rather than increased treatment complexity alone.
Comment 5. Missing Long-Term Outcomes:
- The study focuses on 30-day survival but does not include long-term follow-up or quality-of-life outcomes. These aspects are essential for understanding the true benefit of TPE in ALF.
Author Response: We appreciate this important observation. The primary objective of our study was to evaluate short-term survival, specifically 30-day mortality, during the acute phase of ICU management for ALF. This timeframe was chosen because early mortality remains the most critical determinant of outcome in this patient population, particularly in non-transplant candidates. Unfortunately, long-term follow-up and quality-of-life data were not available for this retrospective cohort. We agree that these outcomes are essential to fully assess the long-term benefits of TPE and have noted this as a limitation in the revised Discussion section. We also emphasize the need for future prospective studies that include long-term survival and functional outcomes.
Comment 6. Minor Comments:
- Language and Grammar:
- Minor grammatical edits are needed throughout. For example, replace “noncetaminophen” with “non-acetaminophen” (line 170).
- Use consistent terminology (e.g., “SMT group” vs. “standard medical treatment group”).
- Figures and Tables:
- Improve clarity of Figure 1 (Kaplan-Meier curve): consider enlarging axis labels and adding a legend to identify groups clearly.
- Include the total number at risk at specific time points in Kaplan-Meier figures.
- Ethics Statement:
- The informed consent statement should be clearly placed in the Methods or Ethics section rather than embedded at the end.
- Conflict of Interest:
- While a potential COI is declared (one author is employed at Terumo BCT), a stronger statement clarifying that Terumo had no role in study design, analysis, or manuscript preparation would improve transparency.
Author Response. We thank the reviewer for these helpful suggestions and have addressed all the points as follows:
- Language and Grammar:
We have carefully reviewed the manuscript and made minor grammatical corrections throughout. Specifically, “noncetaminophen” has been corrected to “non-acetaminophen,” and terminology has been standardized to consistently use “TPE group” and “SMT group” throughout the text. - Figures and Tables:
Figure 1 (Kaplan–Meier curve) has been updated to improve visual clarity. Axis labels have been enlarged, and a clear legend has been added to distinguish between groups. In addition, we have included the number of patients at risk at specific time points below the Kaplan–Meier plot. - Ethics Statement:
The informed consent statement has been moved to the Methods section under Ethical Approval, where it is now clearly stated in accordance with journal guidelines. - Conflict of Interest:
We have revised the Conflict of Interest section to explicitly state that Terumo BCT had no role in the design of the study; in the collection, analysis, or interpretation of data; or in the writing of the manuscript or the decision to submit it for publication.
We sincerely appreciate the reviewer’s attention to detail, which has helped improve the clarity, transparency, and overall quality of our manuscript.
Reviewer 2 Report
Comments and Suggestions for Authors
Thanks for the opportunity to review this important manuscript .
The topic is very relevant due to the high mortality rate from acute liver failure in developed and undeveloped countries. Also informative is the prevalence of this disease in Mexico and the ethnic characteristics of response of Mexican people to standard pharmacotherapy and plasma exchange in treatment of this disease.
Please add "In Mexico" to the title.
The introduction is too short. Please add more information about similar studies in other countries and whether there are any in neighboring countries. It would be interesting if you compared statistics by year.
The purpose requires a more precise formulation.
The paragraph between lines 98-110 needs to be shortened. Why are these details necessary? “The laboratory tests included the following: coagulation studies: PT, INR, activated prothrombin time; complete chemistry panel:Na, K, Cl, Ca, Mg, P, HCO3, glucose, total and direct bilirubin, gamma-glutamyl transferase, alkaline phosphatase, ASAT, ALAT, albumin, creatinine, blood urea nitrogen, arterial blood gas, complete blood count, blood group test, serum acetaminophen levels and toxicology screening; and hepatotropic virus serology: Immunoglobulin M antibody (IgM) to Hepatitis A Virus, Hepatitis B surface antigen, IgM antibody to Hepatitis B core antigen, Antibody to Hepatitis E Virus, Antibody to Hepatitis C Virus, Hepatitis C Virus Ribonucleic Acid, IgM antibody to Herpes Simplex Virus type 1, Varicella Zoster Virus, and ceruloplasmin levels (for Wilson’s disease); pregnancy test; ammonia levels; autoimmune markers: Antinuclear Antibodies, Anti-Smooth Muscle Antibodies and serum immunoglobulin levels; and Human Immunodeficiency Virus 1 and 1, amylase, and lipase. Imaging studies include ultrasound and angiography to rule out tumors or Budd‒Chiari syndrome, although these results are not always definitive.
Please describe the design of your study.
Please indicate what were the exclusion criteria in your study. How many patients were excluded and for what reasons?
What methodology was used to determine the sample size of your study?
Please indicate the intervention section by points
This paragraph between lines 130-134 does not belong in the intervention section. “We confirm that this study was conducted in accordance with the principles outlined in the Declaration of Helsinki (1975), as revised in 2013. Ethical approval was obtained from the Institutional Review Board of Hospital Juárez, Mexico, on October 12, 2023 (approval reference number HJM 071/24-R). Written informed consent was obtained from all participants for inclusion in the study and for the publication of this manuscript.”
By what criteria did some patients undergo SMT and others plasma exchange?
The description of table 1 contains many explanations which are best placed for them in the statistics section. In addition, the table description does not comply with the journal rules.
The results are unclear. Please indicate how many patients were included in the different groups, divided according to the degree of encephalopathy.
In general, the result needs to be redone. Only tables and figures without sufficient description and explanation.
The discussion is written on the basis of the data obtained, with a comparison with the results obtained by other authors.
The conclusions are short and highly related to the results obtained. However in will be more informative if was added "in Mexico" to conclusion.
Author Response
Comment 1. The topic is very relevant due to the high mortality rate from acute liver failure in developed and undeveloped countries. Also informative is the prevalence of this disease in Mexico and the ethnic characteristics of response of Mexican people to standard pharmacotherapy and plasma exchange in treatment of this disease.
Please add "In Mexico" to the title.
Authors respond. Thank you for your comment. It was added
Comment 2. The introduction is too short. Please add more information about similar studies in other countries and whether there are any in neighboring countries. It would be interesting if you compared statistics by year.
Authors’ response: Thank you for highlighting the need for a broader regional context. We have substantially expanded the Introduction to include epidemiological data and key studies from neighboring Latin‑American countries.
Comment 3. The purpose requires a more precise formulation.
Authors respond: Thank you for this helpful suggestion. We agree that a clearer and more precise formulation of the study objective enhances the overall quality of the manuscript. We have revised the final paragraph of the Introduction accordingly
Comment 4. The paragraph between lines 98-110 needs to be shortened. Why are these details necessary? “
The laboratory tests included the following: coagulation studies: PT, INR, activated prothrombin time; complete chemistry panel:Na, K, Cl, Ca, Mg, P, HCO3, glucose, total and direct bilirubin, gamma-glutamyl transferase, alkaline phosphatase, ASAT, ALAT, albumin, creatinine, blood urea nitrogen, arterial blood gas, complete blood count, blood group test, serum acetaminophen levels and toxicology screening; and hepatotropic virus serology: Immunoglobulin M antibody (IgM) to Hepatitis A Virus, Hepatitis B surface antigen, IgM antibody to Hepatitis B core antigen, Antibody to Hepatitis E Virus, Antibody to Hepatitis C Virus, Hepatitis C Virus Ribonucleic Acid, IgM antibody to Herpes Simplex Virus type 1, Varicella Zoster Virus, and ceruloplasmin levels (for Wilson’s disease); pregnancy test; ammonia levels; autoimmune markers: Antinuclear Antibodies, Anti-Smooth Muscle Antibodies and serum immunoglobulin levels; and Human Immunodeficiency Virus 1 and 1, amylase, and lipase. Imaging studies include ultrasound and angiography to rule out tumors or Budd‒Chiari syndrome, although these results are not always definitive.
Authors’ response: Thank you for this valuable observation. We agree that the paragraph was overly detailed and may have detracted from the clarity of the Methods section. In response, we have revised the text to summarize the key diagnostic and exclusion tests used for patient eligibility, grouping them into clinically relevant categories. This preserves transparency and reproducibility while improving readability. We appreciate your suggestion, which helped streamline the manuscript.
Comment 5. Please describe the design of your study.
Authors’ response: Thank you for this important comment. We have clarified the study design in the Methods section. Specifically, we now describe the study as a retrospective, single-center, observational cohort study comparing outcomes in ALF patients treated before and after the implementation of TPE at our institution. We also explain the non-randomized, sequential nature of patient allocation, reflecting the real-world introduction of this intervention in our ICU.
Comment 6.Please indicate what were the exclusion criteria in your study. How many patients were excluded and for what reasons?
Authors’ response: Thank you for your observation. We have added a detailed description of the exclusion criteria and the number of patients excluded in the Eligibility subsection of the Methods (manuscript p. 4, lines 102–106). Specifically, patients were excluded if they were pregnant, in the immediate or late postpartum period, had acute-on-chronic liver failure, withdrew consent, were transferred to another hospital, or had incomplete clinical data.
Comment 7. What methodology was used to determine the sample size of your study?
Authors’ response: Thank you for this important question. As this was a retrospective observational study, the sample size was determined by convenience, based on the total number of patients with a confirmed diagnosis of acute liver failure (ALF) admitted to our ICU between 2018 and 2024 who met the eligibility criteria. No formal statistical power calculation was performed in advance, as our objective was to describe real-world outcomes from all eligible cases treated during this time period.
Comment 8. Please indicate the intervention section by points
This paragraph between lines 130-134 does not belong in the intervention section. “We confirm that this study was conducted in accordance with the principles outlined in the Declaration of Helsinki (1975), as revised in 2013. Ethical approval was obtained from the Institutional Review Board of Hospital Juárez, Mexico, on October 12, 2023 (approval reference number HJM 071/24-R). Written informed consent was obtained from all participants for inclusion in the study and for the publication of this manuscript.”
Authors’ response. Thank you for your constructive feedback. We agree to remove Ethics Statement from the Intervention section.
Comment 9. By what criteria did some patients undergo SMT and others plasma exchange?
Authors’ response: Thank you for this important question. The treatment allocation in our study was based on a sequential cohort design. Patients in the SMT group were treated between 2018 and 2021, prior to the implementation of therapeutic plasma exchange (TPE) at our ICU. Once TPE became available in 2021, it was offered as part of routine care for eligible ALF patients, and these patients formed the TPE group (treated between 2021 and 2024). Thus, the distinction between SMT and TPE groups reflects temporal availability of TPE rather than clinical selection or randomization. We have clarified this point in the Study Design and Population subsection of the Methods.
Comment 10. The description of table 1 contains many explanations which are best placed for them in the statistics section. In addition, the table description does not comply with the journal rules.
Authors’ response: Thank you for your helpful observation. In response, we have simplified the legend of Table 1 to focus on the essential information and relocated the explanation of statistical tests to the Statistical Analysis section of the Methods, in accordance with journal formatting guidelines. We have also ensured that the revised table legend complies with MDPI’s formatting standards.
Comment 11. The results are unclear. Please indicate how many patients were included in the different groups, divided according to the degree of encephalopathy.
In general, the result needs to be redone. Only tables and figures without sufficient description and explanation.
Authors’ response:
Thank you for your constructive feedback. We agree that greater clarity and narrative detail were needed in the Results section. In response, we have thoroughly revised this section to provide a more comprehensive and structured description of our findings.
Specifically:
- We have added a clear breakdown of the number of patients in each treatment group (SMT and TPE), stratified by the degree of hepatic encephalopathy (Grades 0–4), as requested.
- We expanded the accompanying text to describe and interpret the key findings shown in the tables and figures, including survival analysis, regression modeling, and baseline differences between groups.
- Each result is now contextualized with a short summary of its clinical significance and, where applicable, statistical significance.
Comment 12. The discussion is written on the basis of the data obtained, with a comparison with the results obtained by other authors. The conclusions are short and highly related to the results obtained. However in will be more informative if was added "in Mexico" to conclusion.
Authors’ response: Thank you for this thoughtful suggestion. We agree that specifying the geographic context adds clarity and relevance to the conclusion. Accordingly, we have revised the first sentence of the Conclusion section to include the phrase “in Mexico”.
Reviewer 3 Report
Comments and Suggestions for Authors
I have the following concerns:
- Was the effect of RBC transfusions evaluated in all the patients? As far as i read, SMT (standard medical treatment) included antibiotics (piperacillin and tazobactam), N-acetylcysteine, and anti-cerebral edema treatment. Carbapenem and antifungal agents (fluconazole) were initiated for patients whose clinical status worsened, with or without elevated procalcitonin. There is no information regarding RBCs. Were RBCs also administered in high and standard volume TPE? Please answer to the results, if there is an important result regarding RBCs or in the discussion, if it was not conducted.
- TPE was done with cryoprecipitate or with cryosupernatant or both as the replacement fluid? Please answer that to the materials and methods.
- Please expand more the restrictions and limitations of your study in the discussion.
- Was the number of patients studied sufficient in all subgroups to reach safe conclusions? Do guidelines derive from your study?
Author Response
Comment 1. Was the effect of RBC transfusions evaluated in all the patients? As far as i read, SMT (standard medical treatment) included antibiotics (piperacillin and tazobactam), N-acetylcysteine, and anti-cerebral edema treatment. Carbapenem and antifungal agents (fluconazole) were initiated for patients whose clinical status worsened, with or without elevated procalcitonin. There is no information regarding RBCs. Were RBCs also administered in high and standard volume TPE? Please answer to the results, if there is an important result regarding RBCs or in the discussion, if it was not conducted.
Authors’ response:
Thank you for this insightful question. We confirm that red blood cell (RBC) transfusions were not systematically recorded or analyzed.
In general, RBC transfusions are not routinely required during therapeutic plasma exchange (TPE), as the procedure primarily removes plasma and does not deplete red blood cells. Replacement fluids—typically 20% albumin and/or fresh frozen plasma—are used to maintain volume and hemodynamic stability, and RBCs are preserved in the patient’s circulation.
RBC transfusions are only considered in specific clinical situations, such as:
- Pre-existing significant anemia (e.g., Hb <7–8 g/dL with symptoms),
- Unexpected blood loss or hemolysis (both rare in modern TPE systems),
- Progressive anemia after multiple TPE sessions.
Comment 2. TPE was done with cryoprecipitate or with cryosupernatant or both as the replacement fluid? Please answer that to the materials and methods.
Authors’ response: Thank you for your question. In our study, therapeutic plasma exchange (TPE) was performed exclusively using fresh frozen plasma (FFP) from volunteer donors as the replacement fluid. Neither cryoprecipitate nor cryosupernatant was used. At our institution, FFP is the only plasma product routinely available and approved for use in TPE. We have added this clarification to the Materials and Methods section.
Comment 3. Please expand more the restrictions and limitations of your study in the discussion.
Authors´s response. Thank you for this important suggestion. We agree that a more detailed discussion of the study's limitations provides greater transparency and helps frame the interpretation of our findings. In response, we have expanded the Limitations subsection of the Discussion
Comment 4. Was the number of patients studied sufficient in all subgroups to reach safe conclusions? Do guidelines derive from your study?
Authors’ Response: Thank you for this important and thoughtful observation. We agree that the limited sample size, particularly within subgroups (e.g., high- vs. standard-volume TPE and encephalopathy grades), restricts the ability to draw definitive or generalizable conclusions. The primary objective of our study was to provide real-world observational data from a tertiary ICU in Mexico, rather than to perform a powered subgroup analysis or develop clinical guidelines.
Therefore, while our findings suggest a potential survival benefit of therapeutic plasma exchange (TPE) in acute liver failure, we recognize that these results must be interpreted within the limitations of the study design and sample size.
Round 2
Reviewer 1 Report
Comments and Suggestions for Authors
Strengths:
-
Clinical Relevance:
-
ALF carries high mortality, and access to liver transplantation is limited in many parts of the world, including Latin America. This study addresses a significant gap in regional data regarding TPE as a bridge or alternative to transplantation.
-
-
Real-World Setting:
-
The study design reflects actual clinical practice and resource availability, enhancing the external validity for similar healthcare settings.
-
-
Clear Outcomes:
-
The primary endpoint (30-day survival) is clinically meaningful and aligns with prior studies on ALF management.
-
-
Statistical Rigor:
-
Kaplan–Meier survival analysis and binomial regression adjusted for encephalopathy grade support the robustness of the findings, despite the small sample size.
-
-
Well-Described Methodology:
-
Detailed eligibility criteria, intervention protocol (including plasma exchange volumes and replacement fluids), and statistical analysis enhance transparency and reproducibility.
-
Major Concerns:
-
Sample Size and Power:
-
The total sample size (n=25) is limited, and the analysis combining SV-TPE and HV-TPE reduces granularity. Although justifiable, this compromises subgroup insights. A larger cohort or multicenter data would be ideal for confirmatory evidence.
-
-
Temporal Bias:
-
The two groups were treated in different time frames (pre- vs. post-TPE availability). Changes in supportive care practices or ICU staffing over time could confound results. This limitation is acknowledged but warrants emphasis as a significant bias.
-
-
Baseline Imbalance:
-
The TPE group had significantly higher encephalopathy grades at baseline (mean 3.4 vs. 1.7, p=0.005), which could have skewed expected outcomes. While this imbalance makes the survival benefit of TPE more compelling, it also introduces complexity in interpretation.
-
Minor Comments and Suggestions:
-
Clarify Ethical Statement Duplication:
-
Ethical approval and informed consent are mentioned twice (lines 112–116 and 181–185). Consolidate to avoid redundancy.
-
-
Figure/Table Clarity:
-
Ensure all tables and figures (e.g., Figure 1, Table 1) are adequately labeled, referenced in the text, and included in the final submission version with proper resolution.
-
-
Grammar and Formatting:
-
A few typographical issues and minor grammatical inconsistencies are present (e.g., “Standar” instead of “Standard” in Table 2). A thorough proofreading is advised.
-
-
TPE Volume Protocols:
-
Although grouped together, consider briefly discussing whether SV-TPE appeared comparably effective to HV-TPE in descriptive terms (even if not statistically powered for this).
-
-
Long-Term Outcomes:
-
While the focus on 30-day survival is valid, the absence of long-term outcomes (neurological sequelae, quality of life, transplant-free survival) limits the scope of clinical implications. This should be highlighted as a key area for future research.
-
minor correction for quality of English
Author Response
Minor Comments and Suggestions:
- Clarify Ethical Statement Duplication:
- Ethical approval and informed consent are mentioned twice (lines 112–116 and 181–185). Consolidate to avoid redundancy.
Authors’ Response:
We thank the reviewer for pointing out the redundancy. The repeated statement regarding ethical approval and informed consent has been removed from lines 181–185 to improve clarity and avoid duplication.
- Figure/Table Clarity:
- Ensure all tables and figures (e.g., Figure 1, Table 1) are adequately labeled, referenced in the text, and included in the final submission version with proper resolution.
Authors’ Response:
Thank you for your observation. We have ensured that all figures and tables are correctly labeled, referenced in the text, and included in the final submission. The figures have been updated to meet the required resolution standards, and the lettering has been amplified to enhance clarity and readability.
- Grammar and Formatting:
- A few typographical issues and minor grammatical inconsistencies are present (e.g., “Standar” instead of “Standard” in Table 2). A thorough proofreading is advised.
Authors’ Response:
We appreciate the reviewer’s suggestion. The manuscript has been thoroughly revised by the MDPI Author Services editorial team to address typographical and grammatical issues, including the correction of “Standar” to “Standard” in Table 2.
Authors’ Response: We appreciate the reviewer’s suggestion. The manuscript has been thoroughly revised by the MDPI Author Services editorial team to address typographical and grammatical issues, including the correction of “Standar” to “Standard” in Table 2.
- TPE Volume Protocols:
- Although grouped together, consider briefly discussing whether SV-TPE appeared comparably effective to HV-TPE in descriptive terms (even if not statistically powered for this).
Authors’ Response:
Thank you for the suggestion. As recommended, we have added a brief comparison of SV-TPE and HV-TPE outcomes in the Discussion section, highlighting observed trends while acknowledging the limitations due to sample size and lack of statistical power.
- Long-Term Outcomes:
- While the focus on 30-day survival is valid, the absence of long-term outcomes (neurological sequelae, quality of life, transplant-free survival) limits the scope of clinical implications. This should be highlighted as a key area for future research.
Authors’ Response:
We appreciate the reviewer’s insightful comment. We agree that the absence of long-term outcome data, such as neurological sequelae, quality of life, and transplant-free survival, limits the broader clinical implications of our findings. This has now been acknowledged in the Discussion section as an important limitation and a key area for future research to better understand the sustained impact of TPE in this patient population.
Reviewer 2 Report
Comments and Suggestions for Authors
I am very grateful for the opportunity to review this manuscript a gain.
The title in the revised version is more in line with the content.
The added fragments in introduction, especially the epidemiological part, strengthen this section.
The purpose in revised version has become more specific.
The change in materials and methods brought greater clarity to the work being carried out.
The changes made to the results with numerous explanations and clarifications increased the significance and reliability of the results obtained.
I think adding the words "in Mexico" to the conclusions made the conclusions more precise and tied them more closely to the results obtained, which are closely related to the regions of observation.
Author Response
Authors’ Response: Thank you so much for helping us improve our work.
Reviewer 3 Report
Comments and Suggestions for Authors
I have no further concerns.
Author Response

(The authors gave the same response as above.)
